Extended Abstract Track

# Equivariant Representations for Non-Free Group Actions

**Luis Armando Pérez Rey**[*1,2,3]                                L.A.PEREZ.REY@TUE.NL
**Giovanni Luca Marchetti**[*4]                                        GLMA@KTH.SE
**Danica Kragic**[4]                                                          DANI@KTH.SE
**Dmitri Jarnikov**[1,3]                                          D.S.JARNIKOV@TUE.NL
**Mike Holenderski**[1]                                      M.HOLENDERSKI@TUE.NL

*1. Eindhoven University of Technology, Eindhoven, The Netherlands*
*2. Eindhoven Artificial Intelligence Systems Institute (EAISI), Eindhoven, The Netherlands*
*3. Prosus, Amsterdam, The Netherlands*
*4. Royal Institute of Technology (KTH), Stockholm, Sweden*

**Editors:** Sophia Sanborn, Christian Shewmake, Simone Azeglio, Arianna Di Bernardo, Nina Miolane

## Abstract

We introduce a method for learning representations that are equivariant with respect to general group actions over data. Differently from existing equivariant representation learners, our method is suitable for actions that are not free i.e., that stabilize data via nontrivial symmetries. Our method is grounded in the orbit-stabilizer theorem from group theory, which guarantees that an ideal learner infers an isomorphic representation. Finally, we provide an empirical investigation on image datasets with rotational symmetries and show that taking stabilizers into account improves the quality of the representations.

**Keywords:** Representation Learning, Symmetries, Group Theory

## 1. Introduction

The problem of incorporating symmetries into representations defines a fundamental challenge and has been considered in a number of recent works (Quessard et al., 2020; Higgins et al., 2022; Cohen and Welling, 2014; Tonnaer et al., 2022; Ahuja et al., 2021). The overall aim is to design representations which preserve symmetries – a property known as *equivariance*. This is because preservation of symmetries leads to the extraction of geometric and semantic structure in data, which can be exploited for reasoning, efficiency and generalization (Bengio et al., 2013). As an example, the challenge of *disentangling* semantic factors of variations has been rephrased in terms of equivariant representations (Higgins et al., 2018; Caselles-Dupré et al., 2019).

The majority of literature relies on the assumption that the group of symmetries acts *freely* on data (Marchetti et al., 2022) i.e., that no datapoint is stabilized by (nontrivial) symmetries. This avoids the need to model stabilizers, which are unknown subgroups of the symmetry group considered. However, non-free group actions arise in several practical scenarios. This happens for example when considering images of

Figure 1

---

objects acted upon by the rotation group via change of orientation. Such objects might be symmetrical, resulting in rotations leaving the image (almost) identical and consequently ambiguous in its orientation (see Figure 1).

In this work we propose a method for learning equivariant representation for general and potentially non-free group actions. Based on the orbit-stabilizer theorem from group theory, we design a model that outputs cosets of the stabilizer subgroup. The representation learner optimizes an equivariance loss based on supervision from symmetries alone. The above-mentioned theoretical results guarantee that a learner infers representations that are isomorphic to the original dataset.

## 2. Group Theory Background

Let $G$ be the group of symmetries with multiplication denoted by $(g, h) \to gh$ and identity denoted by $1 \in G$. Suppose that $G$ acts on a set $\mathcal{X}$ via $(g, x) \to g \cdot x$. The action defines a set of *orbits* $\mathcal{X}/G$ given by the equivalence classes of the relation $x \sim y$ iff $y = g \cdot x$ for some $g \in G$. For each $x \in \mathcal{X}$, the *stabilizer* subgroup is defined as $G_x = \{g \in G \mid g \cdot x = x\}$. Stabilizers are conjugate as $x$ varies in its orbit, and by abuse of notation we refer to the conjugacy class $G_O$ for $O \in \mathcal{X}/G$. The action is said to be *free* if $G_O = \{1\}$ for every $O$.

Recall that a map $\varphi : \mathcal{X} \to \mathcal{Z}$ between sets acted upon by $G$ is said to be *equivariant* if $\varphi(g \cdot x) = g \cdot \varphi(x)$ for every $x \in \mathcal{X}$ and $g \in G$. An equivariant bijection is referred to as *isomorphism*. The following is the fundamental result on group actions (Rotman, 2012).

**Theorem 1 (Orbit-Stabilizer)** *Each orbit $O$ is isomorphic to the set of (left) cosets $G/G_O = \{gG_O \mid g \in G\}$. In other words, there is an isomorphism:*

$$\mathcal{X} \simeq \coprod_{O \in \mathcal{X}/G} G/G_O \quad \subseteq 2^G \times \mathcal{X}/G \tag{1}$$

*where $2^G$ denotes the power-set of $G$ on which $G$ acts by left multiplication $g \cdot A = \{ga \mid a \in A\}$. Moreover, any equivariant map $\varphi : \mathcal{X} \to \coprod_{O \in \mathcal{X}/G} G/G_O$ which induces a bijection on orbits is an isomorphism.*

## 3. Equivariant Representation Learning

Our goal is to design an equivariant representation learner based on Theorem 1. We aim to train a model $\varphi : \mathcal{X} \to \mathcal{Z}$ with a latent space $\mathcal{Z}$ on a loss encouraging equivariance. While we assume that $G$ is known a priori, its action on $\mathcal{X}$ is not and has to be conveyed through data. The ideal choice for $\mathcal{Z}$ is given by $\coprod_{O \in \mathcal{X}/G} G/G_O$ since the latter is the isomorphic to $\mathcal{X}$ (Theorem 1). In other words, $\varphi$ ideally outputs cosets of stabilizers of the inputs. However, the stabilizers are unknown a priori since they depend on the group action. In order to circumvent the modeling of stabilizers and their cosets, we appeal to the following simple result (a proof is provided in the Appendix):

**Proposition 2** *Let $\varphi : \mathcal{X} \to 2^G$ be an equivariant map. Then for each $x \in \mathcal{X}$ from an orbit $O$, $\varphi(x)$ contains a coset of (a conjugate of) $G_O$.*

# Extended Abstract Track

Proposition 2 enables $\varphi$ to output subsets of $G$ instead of cosets of stabilizers. As long as those subsets are *minimal* w.r.t. to inclusion, they will coincide with the desired cosets. Based on this, we define the latent space as $\mathcal{Z} = \mathcal{Z}_G \times \mathcal{Z}_O$ and implement the map $\varphi$ as a pair of neural networks $\varphi_G : \mathcal{X} \to \mathcal{Z}_G$, $\varphi_O : \mathcal{X} \to \mathcal{Z}_O$. The component $\mathcal{Z}_G$ represents cosets of stabilizers while $\mathcal{Z}_O$ represents orbits. Since the output space of a neural network is a finite-dimensional vector space, we assume that $G$ is a linear Lie group, i.e. $G$ is a manifold of matrices, and that the stabilizers of the action are finite. The model $\varphi_G$ first outputs $N$ elements $(\varphi_G^1(x), \cdots, \varphi_G^N(x)) = \varphi_G(x)$ in the matrix Lie algebra $\mathfrak{g}$ that are converted to $G$ by the exponential map $\exp : \mathfrak{g} \to G$. The hyper-parameter $N$ ideally should be chosen to be larger than the cardinality of the stabilizers. On the other hand, the output of $\varphi_O$ consists of a vector of arbitrary dimensionality. The only requirement is that the output space of $\varphi_O$ should have enough capacity to contain $\mathcal{X}/G$.

Our dataset $\mathcal{D}$ consists of samples from the (unknown) group action, meaning that datapoints are triplets $(x, g, y) \in \mathcal{X} \times G \times \mathcal{X}$ with $y = g \cdot x$. Given a datapoint $(x, g, y) \in \mathcal{D}$ the learner $\varphi_G$ optimizes the equivariance loss over its parameters:

$$\mathcal{L}_G(x, g, y) = d(g \cdot \varphi_G(x), \ \varphi_G(y)) \tag{2}$$

where $d$ is a (semi) metric for sets. We opt for the (asymmetric) *Chamfer distance* $d(A, B) = \frac{1}{|A|} \sum_{a \in A} \min_{b \in B} d_G(a, b)$ because of its differentiability properties. Here $d_G$ is a metric on $G$ and is typically set as the (squared) Euclidean one for $G = \mathbb{R}^n$ and as the (squared) Frobenius one for $G = \mathrm{SO}(n)$. As previously discussed we wish $\varphi_G(x)$, when seen as a set, to be minimal in cardinality. To this end we add the following regularization term measuring the discrete entropy:

$$\widetilde{\mathcal{L}}_G(x) = \frac{\lambda}{N^2} \sum_{1 \leq i,j \leq N} d_G(\varphi_G^i(x), \ \varphi_G^j(x)) \tag{3}$$

where $\lambda$ is a small weight set canonically to 0.001. On the other hand, since orbits are invariant to the group action $\varphi_O$ optimizes a *contrastive loss*. We opt for the popular InfoNCE loss from the literature (Chen et al., 2020):

$$\mathcal{L}_O(x, y) = d_O(\varphi_O(x), \ \varphi_O(y)) - \log \mathbb{E}_{x'} \left[ e^{-d_O(\varphi_O(x'), \ \varphi_O(x))} \right] \tag{4}$$

where $x'$ is marginalized from $\mathcal{D}$. As customary for the InfoNCE loss, we normalize the output of $\varphi_O$ and set $d_O(a, b) = -\cos(\angle ab) = -a \cdot b$. The second summand of $\mathcal{L}_O$ encourages injectivity of $\varphi_O$ as and prevents orbits from collapsing in the representation.

The Orbit-Stabilizer Theorem guarantees that an ideal learner achieves isomorphic representations in the following sense. If the $\mathcal{L}_G(x, g, y)$ and the first summand of $\mathcal{L}_O(x, y)$ vanish for every $(x, g, y)$ then $\varphi$ is equivariant. If moreover the regularizations ($\widetilde{\mathcal{L}}_G$ and the second summand of $\mathcal{L}_O$) are at a minimum then $\varphi_G(x)$ coincides with a coset of $G_O$ for every $x \in O$ (Proposition 2) and $\varphi_O$ is injective. Theorem 1 implies then that the representation is isomorphic (on its image) as desired.

## 4. Experiments

We test an implementation of the neural networks $\varphi_G$ and $\varphi_O$ on the following four datasets consisting of $64 \times 64$ images subject to non-free group actions:

- ROTATING ARROWS: images of radial configurations of $\nu \in \{4, 5, 6\}$ arrows rotated by $G = \mathrm{SO}(2)$. The number of arrows $\nu$ determines the orbit with stabilizer the cyclic group $C_\nu \subseteq G$ of order $\nu$.

- DOUBLE ARROWS: images of two radial configurations of 2 and 3 arrows respectively rotated by $G = \mathrm{SO}(2) \times \mathrm{SO}(2)$. The action produces a single orbit, i.e. it is *transitive*, and the stabilizer is a product of cyclic groups $C_2 \times C_3$.

- CHAIR: images of a monochromatic chair from ModelNet40 (Wu et al., 2015) rotated by $G = \mathrm{SO}(2)$ along a vertical axis with a single orbit with stabilizer the cyclic group $C_4 \subseteq G$ of order 4.

- TETRAHEDRON: images of a monochromatic tetrahedron (Murphy et al., 2021) rotated by $G = \mathrm{SO}(3)$ with a single orbit and stabilizer the alternating group $A_4$ of order 12.

We compare our model with the *baseline* where $\varphi_G$ produces a single output i.e., $N = 1$. The latent space is thus $\mathcal{Z} = G \times \mathcal{Z}_O$, on which $G$ acts freely. This is similar to what has been proposed in previous work (Caselles-Dupré et al., 2019; Marchetti et al., 2022; Tonnaer et al., 2022). The models are compared based on two evaluation metrics. First, the equivariance loss (Equation 3) on a test set. Second, the reconstruction loss (pixel-wise cross-entropy) on a test set of a decoder $\psi : \mathcal{Z} \to \mathcal{X}$ trained jointly with $\varphi$. Quantitative results are presented in Table 1 while qualitative visualizations are shown in Figure 2. As can be seen, our model correctly infers the stabilizers (the cyclic subgroups of $\mathrm{SO}(2)$ and the alternating subgroup of $\mathrm{SO}(3)$) by overlapping components of $\varphi_G(x)$ and distributing them geometrically. Moreover, our model achieves significantly lower scores than the baseline. The latter is not able to capture the stabilizers in its latent space, leading to representations of poor quality and loss of information.

Table 1: Mean and std over 3 runs of the evaluation metrics for our model and the baseline.

| Dataset | Model | N | Equivariance | Reconstruction |
|---|---|---|---|---|
| ROTATING ARROWS | Baseline | 1 | $1.985_{\pm 0.027}$ | $0.551_{\pm 0.051}$ |
| | Ours | 10 | $0.011_{\pm 0.002}$ | $0.372_{\pm 0.040}$ |
| DOUBLE ARROWS | Baseline | 1 | $4.016_{\pm 0.027}$ | $0.219_{\pm 0.002}$ |
| | Ours | 6 | $0.009_{\pm 0.006}$ | $0.152_{\pm 0.011}$ |
| CHAIR | Baseline | 1 | $1.944_{\pm 0.045}$ | $0.603_{\pm 0.037}$ |
| | Ours | 5 | $0.098_{\pm 0.061}$ | $0.424_{\pm 0.022}$ |
| TETRAHEDRON | Baseline | 1 | $6.025_{\pm 0.063}$ | $0.470_{\pm 0.045}$ |
| | Ours | 20 | $0.032_{\pm 0.007}$ | $0.286_{\pm 0.029}$ |

## 5. Conclusions and Future Work

In this work we introduced a method for learning equivariant representations for general and potentially non-free group actions. We discussed the theoretical foundations and empirically investigated the method on images with rotational symmetries.

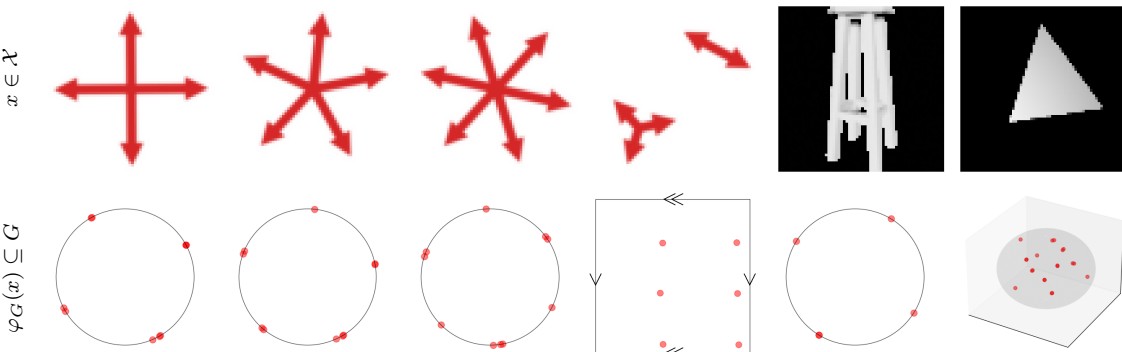

Figure 2: Visualization of the data $x$ for the four datasets and the predicted stabilizer $\varphi_G(x)$. For the double arrows, the torus $G = \mathrm{SO}(2) \times \mathrm{SO}(2)$ is visualized as an identified square. For the tetrahedron, $G$ is visualized as a projective space $\mathbb{RP}^3 \simeq \mathrm{SO}(3)$.

Our model relies on the assumptions that the stabilizers are finite. However, non-discrete stabilizer subgroups sometimes occur, for example in the case of symmetrical objects such as a cone or a cylinder. An interesting future direction is designing an equivariant representation learner suitable for group actions with non-discrete stabilizers and the evaluation of our method on more complex datasets.

## 6. Acknowledgements

The second and third named authors thank the Swedish Research Council, the Knut and Alice Wallenberg Foundation and the European Research Council (ERC-BIRD-884807) for their support. This work has also received funding from the NWO-TTW Programme "Efficient Deep Learning" (EDL) P16-25.

Extended Abstract Track

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

## 7. Appendix

### 7.1. Proofs of Theoretical Results

**Proposition 3** *Let $\varphi : \mathcal{X} \to 2^G$ be an equivariant map. Then for each $x \in \mathcal{X}$ from an orbit $O$, $\varphi(x)$ contains a coset of (a conjugate of) $G_O$.*

**Proof** Pick $x \in \mathcal{X}$. Then for every $g \in G_x$ it holds that $\varphi(x) = \varphi(g \cdot x) = g \cdot \varphi(x)$. In other words $G_x h = h h^{-1} G_x h \subseteq \varphi(x)$ for each $h \in \varphi(x)$. Since $h^{-1} G_x h$ is conjugate to $G_x$ the thesis follows. ∎

### 7.2. Training Details

We implement the neural networks $\varphi_G$ and $\varphi_O$ with a backbone ResNet18 (He et al., 2016). For a datapoint $x \in \mathcal{X}$, the network implements multiple heads to produce embeddings $\left(\varphi_G^1(x), \cdots, \varphi_G^N(x)\right)$ with $\varphi_G^i(x) \in G$. The output dimension of $\varphi_O$ is set to 3. We train the model for 50 epochs (except for the chair datasaet which requires a longer training of 100 epochs) using the AdamW optimizer (Loshchilov and Hutter, 2019) with a learning rate of $10^{-4}$ and batches of 16 triplets $(x, g, y) \in \mathcal{D}$.

The rotating arrows and chair dataset consists of 5000 datapoints per orbit while the double arrows and the tetrahedron datasets consist of 20000 datapoints. For all datasets the test set consists of a random 10% split.

### 7.3. Reconstructions

We present in Figure 3 some examples of images reconstructed by a decoder $\psi$ trained jointly with the encoder $\varphi$. The baseline model is not capable of clearly reconstructing the images compared to our model.

As seen from the quantitative results in Table 1, the baseline model is not capable of optimizing the equivariance loss from Equation 2. This provides a hint that the encoder $\varphi$ might not be converging to a stable representation. Consequently, the decoder is incapable of consistently reconstructing the data which results in a higher reconstruction loss.

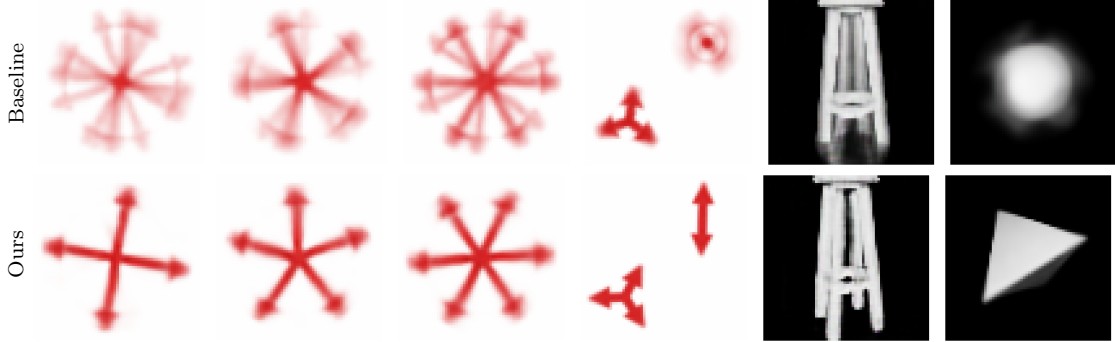

Figure 3: Reconstruction examples produced by a decoder trained on the latent variables produced by our model and the baseline.

