# OpenReview forum: "Equivariant Representations for Non-Free Group Actions"
_NeurIPS.cc/2022/Workshop/NeurReps — NeurReps 2022 Poster_

### Official Review · Reviewer_qW8z · 2022-10-07
**Interesting and theory grounded method for an important problem, but with an inconsistency**

**Confidence:** 4
**Soundness:** 3
**Presentation:** 2
**Contribution:** 3
**Overall Rating:** 4

**Summary:**

The submission describes a group equivariant representation learning method where the symmetry group G is assumed to be known, but its action on the data space is not (3rd sentence of section 3). The embedding is performed by a pair of neural networks that is trained self-supervised using an equivariance loss. Observing that the data space X is by the group action partitioned into group orbits, the authors propose to embed the specific orbit of a data point in X/G via a first, G-invariant network, and the G-pose of the datapoint within the orbit via a second, G-equivariant network. If the action is not fixed point free, i.e. if the datapoint is symmetric under the action of some stabilizer subgroup, the task of the second network is ill defined since the G-pose is only defined up to this symmetry. To solve the issue, the second network predicts a whole set of group elements (G-poses) - Proposition 2 ensures that this set has to be a coset of the stabilizer subgroup if the network is to be equivariant.

The model is trained by presenting it triples (x,g,y), where g is a group element and x,y are data points related by the action of g according to y=g.x. The first, invariant network is optimized using a contrastive loss, which brings the elements x,y in the same orbit close together and pushes orbits on different orbits apart. The second network is optimized to minimize the Chamfer distance between the predicted coset for y and the transformation of the predicted coset for x by g.

To test the method, the authors embed synthetic images with controlled symmetries (stabilizer subgroups). It is shown to perform better than a baseline, which predicts a single (ill-defined) group element (G-pose). The comparison metrics hereby are the equivariance loss and a reconstruction loss, training a decoder for the latter.

**Questions:**

see above

**Limitations:**

The authors discuss the limitation that the method works currently only for finite stabilizer subgroups.

**Recommended Decision:**

2: Borderline

**Relevance:**

3: Solid fit

**Strengths And Weaknesses:**

The method is novel in so far that it addresses symmetries in the data to be embedded, which prior work did not. It is well rooted in the orbit stabilizer theorem and proposition 2 and seems technically sound.

What is unclear to me is how the experiments fit together with the assumption that the group action is unknown, since the training dataset is constructed by creating pairs x and y=g.x that are constructed using this allegedly missing action. This is a severe inconsistency, which can only be resolved by either changing the assumption that the action is unknown, or the training method itself. Due to this inconsistency I am currently reluctant to give a higher overall score than 4 and set the recommendation to borderline.

The presentation could be improved: to see where the authors are heading, the related work and baseline model which assume free group actions could be described/formalized before presenting the solution. As the paper is organized currently, it needs to be read twice before the idea becomes clear.
The network \phi_G is is defined to predict Lie algebra elements, which can via the exponential map be mapped to group elements. However, the losses in equations 2 and 3 seem to miss the exp map?

The problem seems important, however, the significance of the solution is not entirely clear without being tested on more realistic datasets.

**Submission Track:**

Extended Abstract (4 Page)

---

### Official Review · Reviewer_gpb2 · 2022-10-14

**Confidence:** 3
**Soundness:** 3
**Presentation:** 2
**Contribution:** 3
**Overall Rating:** 7

**Summary:**

The paper suggests a method for learning representations that are equivariant to the action of some specified group G. The central contribution is to extend previous methods to the setting of non-free group actions. They do so by decomposing the latent space into two components corresponding to: i) Orbit ii) Coset of stabilizers. They learn two separate neural networks for each of these components. Sec. 4 shows some interesting experiments of arrows and tetrahedrons subject to non-free group action. They show that their method outperforms the baseline which assumes that the action is free.

**Questions:**

In section 3 "Theoretically, the ideal choice for Z is given by the right-hand side of the isomorphism in Equation 1.". This claim could use some further clarification. Perhaps just one sentence on why this is the ideal choice for Z.

Is the converse of proposition 2 true? If for \phi: X -> 2^G, we have that for each x from orbit O,  \phi(x) contains some coset of G_O is \phi an equivariant map?

**Limitations:**

The paper accurately identifies its limitation due to the assumption that the stabilizers are finite.

**Recommended Decision:**

3: Accept

**Relevance:**

4: Highly relevant

**Strengths And Weaknesses:**

Originality: The contributions are significant and somewhat new. Aspects of the contributions exist in prior work. It is a generalization of Marchetti et al. (2022) where the authors also learn equivariant representations by decomposing the latent space. But whereas they assume that the group acts freely this paper generalizes the method to non-free group actions.

Quality: The submission is technically sound. The stated propositions are true and the proofs seem correct.

Clarity: The paper is generally clear and the claims are easy to follow.

Significance: The results are interesting because the premise of all groups acting freely is a strong assumption and it seems unjustified.



**Submission Track:**

Extended Abstract (4 Page)

---

### Official Review · Reviewer_QbG5 · 2022-10-16
**Equivariant Representations for Non-Free Group Actions**

**Confidence:** 4
**Soundness:** 2
**Presentation:** 3
**Contribution:** 3
**Overall Rating:** 5

**Summary:**

This paper utilizes a classical decomposition theorem in group theory as an ansatz to learn equivariant maps for a known linear Lie group acting not necessarily freely on a space.  Additionally, a standard neural network (ResNet) is applied in a reconstruction layer to make sure that in addition to the equivariant properties of the situation, there is enough information in the full feature map to make back the input.  The method is applied to a couple of examples of simple patterns in images to demonstrate that the method produces equivariance that can also be used for reconstruction.


**Questions:**

What are the uniqueness claims of this paper?  Can the method be adapted even if the group is unknown? (eg https://arxiv.org/abs/2209.03416). Is it clear that the method transfers to other datasets without more training given that they have solved one (such as "Arrows")?  Why not have a "baseline" that are other popular method(s) in the field?  Where is a standard problem from a standard dataset?  Is N=20 really starting to bottleneck the approach or is that just the number required for those particular problems?  What about input dimension?


**Limitations:**

One direct limitation is that the group has to be known in advance and arise from a given class, but this is not a dealbreaker.  More limiting is the lack of theoretical clarity on the implications of this approach as well as the generalizability to other examples that have practical benefit.  For instance, what does this method do for Rotated MNIST problems?  Overall, much more could have been done to convince this reviewer of the significance of this approach (versus others).


**Recommended Decision:**

2: Borderline

**Relevance:**

4: Highly relevant

**Strengths And Weaknesses:**

The paper is short and fairly clear about the conceptual strategy.  There are some examples from contrived data which give results suggesting that the main goal of the paper is accomplished with them.

On the other hand, these examples are not as convincing as one would desire.  For instance, numbers are given for Equivariance and Reconstruction scores, but given only these scores, it is impossible to determine how good they are.  The examples of reconstruction have to wait to the appendix, but should be presented in the main text.  Moreover, although we see equivariance scores such as ".011", again it is impossible to determine if they are successful scores that demonstrate the claims of the paper.  In particular, if equivariance is really being discovered by the method, why isn't it demonstrated on another set of data never seen before but with the same group acting?  Or is this a limitation of the method?  It is not clear from the paper.

Finally, several times, including in the abstract, introduction, etc. the authors claim that it is "gauaranteed that an ideal learner infers an isomorphic representation".  Or it is stated that the "above-mentioned theoretical results guarantee that a learner infers representations that are isomorphic to the original dataset", when no such guarantees appear before, nor later.  If "ideal learner" means achieve exactly zero error on the loss functions, and if this implies a uniqueness result, it should at least be stated and explained why this is the case.  Otherwise, if a learner gets close to zero, epsilon say, this sounds like a very difficult mathematics problem to prove that the learner finds a solution close to the original.  see, for instance, the difficulty in such a result for sparse coding / compressed sensing:  https://ieeexplore.ieee.org/document/8805108

In general, more clarity on the main claims of the paper are warranted -- at least statements of what kind of theorems or conjectures are being exposited.


**Submission Track:**

Extended Abstract (4 Page)

---

### Decision · Program_Chairs · 2022-10-21

Accept (Poster)